# PSMA and Choline PET for the Assessment of Response to Therapy and Survival Outcomes in Prostate Cancer Patients: A Systematic Review from the Literature

**DOI:** 10.3390/cancers14071770

**Published:** 2022-03-31

**Authors:** Pierpaolo Alongi, Riccardo Laudicella, Helena Lanzafame, Andrea Farolfi, Paola Mapelli, Maria Picchio, Irene A. Burger, Andrei Iagaru, Fabio Minutoli, Laura Evangelista

**Affiliations:** 1Nuclear Medicine Unit, A.R.N.A.S. Ospedale Civico Di Cristina Benfratelli, 90127 Palermo, Italy; pierpaolo.alongi@arnascivico.it; 2Nuclear Medicine Unit, Fondazione Istituto G.Giglio, 90015 Cefalù, Italy; riccardo.laudicella@unime.it; 3Nuclear Medicine Unit, Department of Biomedical and Dental Sciences and Morpho-Functional Imaging, University of Messina, 98122 Messina, Italy; fabio.minutoli@unime.it; 4Department of Nuclear Medicine, University Hospital Zürich, University of Zurich, 8091 Zurich, Switzerland; irene.burger@ksb.ch; 5Department of Nuclear Medicine, University Hospital Essen, University of Duisburg-Essen, 45147 Essen, Germany; helena.lanzafame@uk-essen.de; 6Division of Nuclear Medicine, IRCCS Azienda Ospedaliero-Universitaria di Bologna, University of Bologna, 40138 Bologna, Italy; andrea.farolfi@aosp.bo.it; 7Nuclear Medicine Department, IRCCS San Raffaele Scientific Institute, Vita-Salute San Raffaele University, 20132 Milan, Italy; mapelli.paola@hsr.it (P.M.); maria.picchio@hsr.it (M.P.); 8Department of Nuclear Medicine, Kantonsspital Baden, 5404 Baden, Switzerland; 9Division of Nuclear Medicine and Molecular Imaging, Department of Radiology, Stanford University, Stanford, CA 94035, USA; aiagaru@stanford.edu; 10Nuclear Medicine Unit, Department of Medicine, University of Padua, 35128 Padova, Italy

**Keywords:** prostate cancer, PSMA, choline, PET, response to therapy, prognosis

## Abstract

**Simple Summary:**

Radiolabeled choline and PSMA PET have been largely tested in the initial staging of prostate cancer and for biochemical recurrence. Moreover, diverse data are now available about their role in the evaluation of response to local and systematic therapies, and their predictive impact on the prognosis, before and after therapy. Therefore, in the present systematic review, we aimed to describe the available data, to summarize the current evidence in these settings of disease.

**Abstract:**

The aims of this systematic review were to (1) assess the utility of PSMA-PET and choline-PET in the assessment of response to systemic and local therapy, and to (2) determine the value of both tracers for the prediction of response to therapy and survival outcomes in prostate cancer. We performed a systematic literature search in PubMed/Scopus/Google Scholar/Cochrane/EMBASE databases (between January 2010 and October 2021) accordingly. The quality of the included studies was evaluated following the “Quality Assessment of Prognostic Accuracy Studies” tool (QUAPAS-2). We selected 40 articles: 23 articles discussed the use of PET imaging with [^68^Ga]PSMA-11 (16 articles/1123 patients) or [^11^C]/[^18^F]Choline (7 articles/356 patients) for the prediction of response to radiotherapy (RT) and survival outcomes. Seven articles (three with [^68^Ga]PSMA-11, three with [^11^C]Choline, one with [^18^F]Choline) assessed the role of PET imaging in the evaluation of response to docetaxel (as neoadjuvant therapy in one study, as first-line therapy in five studies, and as a palliative regimen in one study). Seven papers with radiolabeled [^18^F]Choline PET/CT (*n* = 121 patients) and three with [^68^Ga]PSMA-11 PET (*n* = 87 patients) were selected before and after enzalutamide/abiraterone acetate. Finally, [^18^F]Choline and [^68^Ga]PSMA-11 PET/CT as gatekeepers for the treatment of metastatic prostate cancer with Radium-223 were assessed in three papers. In conclusion, in patients undergoing RT, radiolabeled choline and [^68^Ga]PSMA-11 have an important prognostic role. In the case of systemic therapies, the role of such new-generation imaging techniques is still controversial without sufficient data, thus requiring additional in this scenario.

## 1. Introduction

The use of biomarkers is increasingly important in oncology during the initial diagnosis, follow-up evaluation, and for the response to therapy assessment. However, the integration of biomarkers into clinical workflows requires validation in large data sets and endorsement in clinical guidelines. Currently, various agents are available for the treatment of patients with advanced prostate cancer (PCa); therefore, the early detection of response or progression to each therapeutical approach can significantly impact patients’ management, life expectancy, and health costs.

The assessment of local or distant recurrence after radical therapy in PCa patients is currently a topic of discussion and debate in the multidisciplinary scientific community. Prostate Cancer Clinical Trials Working Group 3 (PCWG3), National Cancer Comprehensive Network (NCCN), and European Association of Urology (EAU) guidelines established the use of different diagnostic procedures for this purpose.

Prostate-specific antigen (PSA) is still the most useful biochemical parameter in PCa clinical practice. It has been associated with the tumor burden [1] and it is included in many prognostic tools [2]. However, in patients who developed a castration-resistant condition, PSA cannot be considered the best indicator of response to therapy for several reasons: (1) the increase in disease heterogeneity, (2) the presence of the flare phenomena [3], and (3) the potential development of de-differentiated metastasis not producing PSA [4].

Response Evaluation Criteria in Solid Tumors (RECIST) 1.1 are currently the standard criteria for the definition of response to therapy for soft tissue lesions in patients with hormone-sensitive and castration-resistant PCa (CRPCa) [5]. However, such criteria are limited when evaluating treatment response in bone lesions. Additionally, whole-body magnetic resonance imaging (WB-MRI) has been used for this scope [6], demonstrating some limitations, mainly in the interpretation (in terms of inter-reader agreement) and availability.

Radiolabeled choline and prostate-specific membrane antigen (PSMA) positron emission tomography (PET), associated with computed tomography (CT) or MRI, may play important roles as diagnostic modalities in patients with PCa. Indeed, PSMA- and choline PET/CT or PET/MRI may be useful at any stage of the disease stage, and also in the response to systemic therapies (e.g., docetaxel, abiraterone acetate, enzalutamide, and so on) or local therapies (i.e., external beam radiation therapy (EBRT), stereotactic body radiation therapy (SBRT), salvage lymphadenectomy, and so on) assessment in both hormone-sensitive and CRPC patients. In 2017, Ceci et al. [7] described the role of choline PET in the evaluation of response to therapy in metastatic CRPCa, showing interesting results; however, overall, there is still little evidence available for response assessment with both choline and PSMA PET in general, and EAU guidelines still recommend the use of MRI or PET only within clinical trials.

Therefore, the aim of this systematic review was to evaluate the role of PSMA- and choline-PET for response assessment in PCa, and their potential role as biomarkers for local radiotherapy and systemic treatment evaluations.

## 2. Materials and Methods

### 2.1. Screening and Data Selection

We performed a literature search in the PubMed/Scopus/Google Scholar/Cochrane/EMBASE databases about the role of PSMA- and choline-PET in the response to therapy assessment of PCa using the following terms: “Choline” or “[^18^F]Choline” or “[^11^C]Choline” or “prostate-Specific Membrane Antigen” or “PSMA” or “[^68^Ga]PSMA-11” or “[^18^F]DCFPyL” or “[^18^F]PSMA” AND “prostate cancer” or “prostatic cancer” AND “PET” or “positron emission tomography” AND “therapy response” or “survival”. The search was performed for articles published between January 2010 and October 2021 according to the following filters: original articles, clinical studies, and written in the English language. Editorials, case reports, and review papers were excluded. The literature search was made by four independent authors (A.F., H.L., R.L., and F.M.). All papers were screened by the title and the content of the abstract. Only for the papers that were inherent to the endpoint of the present review were full texts retrieved to verify the relevance of the data. Moreover, for the enrichment of the collected data, all the references of the selected papers were also checked.

### 2.2. Data Extraction

The following data were extracted from all the selected papers: country of origin, number of patients, the median (range) or mean (+ standard deviation—SD) main values, type of therapy, type of radiopharmaceutical, type of hybrid PET system (PET/CT or PET/MRI), and the timing of PET in respect to the therapeutic approach. In addition, data about the interpretation of PET imaging were collected (i.e., visual or semiquantitative analysis).

### 2.3. Reference Standard for the Evaluation of Response to Therapy

For the evaluation of the response to therapy, data about the standard of reference were recovered, such as PSA, histopathology, clinical information, RECIST 1.1. criteria or their combination.

### 2.4. Quality Assessment

The methodological quality of the studies was judged by two investigators (P.A. and L.E.) using the “Quality Assessment of Prognostic Accuracy Studies” tool (QUAPAS) [8] when data about survival were available.

## 3. Results

A total of 40 articles were selected from 4592 records obtained in the PubMed/Scopus/Google Scholar/Cochrane/EMBASE databases according to our research strategy. Figure 1 shows the papers’ selection workflow, following the PRISMA criteria. The QUAPAS of 40 papers is reported in Figure 2.

As illustrated, the majority of papers were at low risk of bias and applicability. Only the analysis of the potential bias was unclear in many reports. The main characteristics of each paper included in this paper are summarized in Table 1.

The articles were divided per topic such as the prognostic value of RT driven by PET imaging, and the evaluation of response to systemic therapy (i.e., chemotherapy, new hormonal agents, enzalutamide, abiraterone acetate, and/or [^223^Ra]dichloride) were singularly assessed (Figure 3).

### 3.1. Radiotherapy and PET Imaging with PSMA or Choline

A total of 23 articles were selected on the use of PET imaging with [^68^Ga]PSMA-11 (16 articles/1123 patients) or choline (7 articles/356 patients) for the prediction of radiotherapy response and survival outcomes. Among these studies, 6 articles described the prognostic role of PET after salvage RT ([9,10,11,12,13,14,15]), while 12 studies assessed the prognostic use of molecular imaging for the treatment of oligometastatic or oligo-recurrent PCa ([16,17,18,19,20,21,22,23,24,25,26,27]). The remaining studies (*n* = 5) were variably based on the prognostic evaluation of PET-guided RT in case of localized PCa [28], newly diagnosed lymph-node-positive PCa [29], and in a mixed population [30]. Finally, one study focused on the prognostic role of PET/CT-guided focal prostate re-irradiation [31]. Several studies regarding the role of choline and [^68^Ga]PSMA-11 PET/CT-based salvage extended field RT (s-EFRT), salvage-involved field RT (s-IFRT), helical tomotherapy (HTT), or SBRT, confirmed that such treatments are safe and effective in case of bone or nodal relapses, both in terms of disease control and long-term survival. In 2017, Emmett et al. [11] posited that PSMA-based salvage-RT (sRT) improves treatment outcomes in the context of biochemical failure after radical prostatectomy (RP). PSMA PET was an independent factor, and when negative (no pathological PSMA uptake), it predicted a high response and progression-free survival (PFS) after a median follow-up of one year from fossa sRT. Schmidt-Hegemann et al. [12] showed in 129 patients who underwent PSMA PET before sRT for PCa recurrence that PSMA PET-based RT is an efficacy salvage treatment, being able to maintain a period of PSA-free relapse similar to that provided by ADT. Another interesting scenario in PCa is oligometastatic and oligo-recurrent disease in which PET-based guided RT may have prognostic utility for (1) local control of disease, and for (2) the opportunity to postpone ADT. As aforementioned, some papers are now available with choline or PSMA PET imaging for this latter setting, which means in the oligometastastic or oligo-recurrent patients, with PSMA literature radically increasing. However, PCa patient cohorts are heterogeneous, including only hormone-sensitive patients ([21,22,24,27]), only castrate-resistant subjects [22], or both ([19,20,25]). Pasqualetti and co-authors [16], in a group of 46 patients with biochemical recurrence PCa limited up to three lesions, defined the ability of [^18^F]Choline PET/CT to identify patients suitable for SBRT, resulting in a systemic therapy-free survival of 39.1 months. Additionally, Bouman-Wammes et al. [17] and Kalinauskaite et al. [19] showed that SBRT based on PSMA-PET can reach an excellent local control, with a reduction in the PSA level, minimal toxicity, and long survival. The literature showed that the local control of disease after SBRT based on PSMA-PET or metastatic direct therapy (MDT) based on PSMA-PET in patients with oligometastatic or oligo recurrent PCa at 1 year is 100% [25], ranging between 92 and 96% at 2 years ([19,20,21]), and 55% at 3 years [32]. Furthermore, in patients with oligometastatic disease treated with RT based on PSMA-PET, the ADT-free survival reached up to 31 months [22] or 74% after 2 years from the local therapy [24]. Interestingly, in the same setting of patients, the overall survival (OS) was superior to 80% in all cases, after 1, 2, or more than 3 years ([19,25,32]). Narang et al. [30] in a phase I/II study, implemented the planning treatment using PSMA PET/MRI with secondary image fusion based on the CT scan for anatomic coregistration. The authors observed excellent five-year outcomes for locally advanced, node-positive, and bone oligometastatic PCa, using dose-escalation extreme hypofractionated (EH)-SBRT boost to the prostate. Molecular imaging also seems particularly useful for re-irradiation in the pelvic region. In this scenario, a recent prospective study of Bergamin et al. validated the use of PSMA-directed salvage focal re-irradiation to the prostate bed using SBRT, confirming the feasibility and safety with a low rate of toxicity and favorable short-term local and biochemical control in a cohort of 25 patients [31].

Most of the studies in the literature focused on the qualitative assessment of PET imaging, and only a few articles assessed the use of semiquantitative parameters in PCa. Incerti et al. [13] demonstrated how [^11^C]choline PET/CT performed as a guide for HTT on lymph node recurrence can be predictive of survival, observing that the metabolic tumor volume (MTV) with a 60% threshold and extrapelvic disease were the best predictors for tumor response and outcome. Differently, in a study by Koerber et al., the use of only standardized uptake value (SUV) was tested in oligo recurrent PCa patients treated with PSMA-guided radiotherapy. The biochemical response was detected in 83% of the cohort, and a statistically significant decrease in SUV was seen in 81% of irradiated metastases. After a median follow-up of 26 months, the three-year OS and biochemical progression-free survival (bPFS) were 84% and 55%, respectively, while the median time of ADT-free survival was 13.5 months [32]. Finally, Franzese et al. [18] and Baumann et al. [23] assessed the utility of pre- and post-RT choline and PSMA PET/CT, respectively, in 38 and 280 oligometastatic patients. Franzese et al. found that post-treatment choline PET/CT was able to identify most patients with a complete or partial metabolic response [18]; Baumann et al. observed that, in analogy to PERCIST, a decrease in PSMA uptake of 30% was observed in 78% of the lesions [23].

### 3.2. Chemotherapy and PET Imaging with PSMA or Choline

Globally, seven articles related to the role of PET-imaging in the evaluation of response to chemotherapy (all docetaxel; as neoadjuvant therapy in one study [33]; as first-line therapy in five [34,35,36,37,38] and as a palliative regimen in one study [39]; Table 1) were included in this review. In four out of seven papers, radiolabeled choline PET was used. In total, 133 and 100 patients underwent baseline and post-docetaxel radiolabeled choline and [^68^Ga]PSMA-11 PET/CT, respectively. The post-docetaxel scan was performed in different ranges of time, either for the radiolabeled choline or [^68^Ga]PSMA-11. Similarly, also the criteria of response to therapy varied from EORTC to change in SUVs or PERCIST criteria. Scharzenbock et al. [33] did not find any correlation among [^11^C]Choline PET metrics, RECIST, and clinical criteria for the evaluation of response to the first-line docetaxel in patients with mCRPCa. Conversely, the same authors [34] affirmed, in a population of 11 patients, that [^11^C]Choline PET/CT may be useful for monitoring the effects of neoadjuvant chemotherapy in locally advanced and high-risk patients. Ceci et al. [35] showed that in patients with mCRPC undergoing first-line docetaxel, [^11^C]Choline PET/CT was able to identify those with a decrease in PSA during therapy, despite radiological progression. Therefore, based on the radiological data, the clinical effect would be a misclassification of patients with an unfavorable prognosis, although a biochemical response to therapy. Additionally, Quaquarini et al. [36] demonstrated the utility of [^18^F]Choline PET/CT in patients undergoing first-line docetaxel, mainly reporting the utility of PET parameters in the prediction of PFS and OS. Indeed, the authors found that the sum of total lesion activity (TLA) and metabolically active tumor volume (MTV) was significantly higher in patients with a worse prognosis. In the last few years, some articles were published, using [^68^Ga]PSMA-11 for the monitoring of response to docetaxel, with the initial results suggesting that [^68^Ga]PSMA-11 PET/CT might be superior to conventional imaging in the evaluation of response to chemotherapy, either in hormone-sensitive or mCRPCa patients ([37,38]). Finally, the group of Simsek [39], similarly to Quaquarini et al. [36], showed that the total baseline PSMA-derived tumor volume (TV-PSMA) may be predictive of response to first-line docetaxel, thus delineating the opportunity to use it as a reliable tool for survival assessment in mCRPCa patients.

### 3.3. Enzalutamide/Abiraterone and PET Imaging with PSMA or Choline

For the use of PET imaging during second-generation antiandrogen therapy, seven articles were selected. Four studies (121 patients) involved radiolabeled choline PET/CT and three (87 patients) used either [^68^Ga]PSMA-11 or [^18^F]DCFPyL PET/CT before and after enzalutamide or abiraterone, respectively. However, the period from the ongoing therapy or the end of therapy and PET scan was completely different across all the selected studies. Several authors assessed the utility of [^18^F]Choline PET/CT before and after enzalutamide [40,41,42]. The groups of Caffo [40] and Maines [41] showed that baseline SUV_max_ and the reduction in metabolic tumor volume (MTV) on the second [^18^F]Choline PET scan might predict the efficacy of the therapy, respectively. Furthermore, De Giorgi et al. [42] described the potential utility of the combination between decreases in PSA and [^18^F]Choline SUV_max_ for the prediction of PFS in mCRPCa. De Giorgi et al. [43] also evaluated the role of [^18^F]Choline PET/CT in monitoring the response to abiraterone in mCRPCa, showing its promising role for the prediction of clinical outcomes beyond PSA response. 

Data about radiolabeled PSMA in this setting are sparse and heterogeneous: Chen et al. [44] reported the utility of [^68^Ga]PSMA-11 PET/CT before and after abiraterone and hormonal therapy as neoadjuvant treatment in patients with advanced hormone-sensitive PCa. They observed that the change in SUVs before and after treatment was significantly associated with the prognosis at univariate analysis. On the contrary, at multivariate analysis, only post-treatment SUVs were associated with the outcome. The groups of Zukotynski [45] and Plouznikoff [46] assessed the uptake’s variations on PSMA-targeted PET, with [^18^F]DCFPyL and [^68^Ga]PSMA-11, respectively, in mCRPCa treated with abiraterone or enzalutamide. Zukotynski et al. [45] observed that PSMA expression’s reduction on PET/CT were strong predictors of response to enzalutamide and abiraterone; Plouznikoff et al. [46] reported that the patterns of change in PSMA uptake were significantly associated with the time to progression and OS after enzalutamide and abiraterone acetate.

### 3.4. Radium223 and PET Imaging with PSMA or Choline

[^18^F]Choline and [^68^Ga]PSMA-11 PET/CT as gatekeepers for the treatment of metastatic PCa with Radium 223 were assessed in three papers. The groups of Filippi [47] and Garcia Vicente [48] assessed 11 and 40 patients, retrospectively, who underwent PET/CT with [^18^F]Choline before [^223^Ra]dichloride treatment. They stated that the presence of extra-osseous disease at PET/CT was correlated with a poor outcome and also a high SUV_max_ (in at least five sites). In the same setting, Filippi et al. [47] demonstrated that the TLA of the dominant lesion was the only predictor of OS. So far, only Ahmadzadehfar et al. [49] have evaluated the utility of [^68^Ga]PSMA-11 PET/CT for planning [^223^Ra]dichloride therapy in patients with mCRPCa. They enrolled 63 patients, who were submitted to conventional imaging and a bone scan (Group 1) or bone scan and PSMA PET (Group 2) before starting [^223^Ra]dichloride therapy. When the assessment of disease extension was made by using the combination of PSMA PET and bone scan, radionuclide therapy with [^223^Ra]dichloride was more effective, mainly regarding the changes in PSA levels during treatment. Therefore, the strategy of the combination of molecular imaging seems promising for the assessment of disease extension, and thus in predicting the response to alpha-therapy with [^223^Ra]dichloride [49].

The summary of each study is reported in Table 2.

**Table 1 cancers-14-01770-t001:** Main characteristics of the selected studies.

Authors, Ref	Year	Country	N pts	Age Median (Range) or Mean (SD)	Type of Therapy	Tracer	Scanner	Time of PET
Incerti et al. [13]	2015	Italy	68	67 (51–81)	sTomotherapy	[^11^C]Choline	PET/CT	baseline
Pasqualetti et al. [16]	2020	Italy	51	70 (50–81)	SBRT	[^18^F]Choline	PET/CT	baseline
Fodor et al. [9]	2017	Italy	81	68 (51–81)	sTomot herapy	[^11^C]Choline	PET/CT	baseline
Bouman-Wammes et al. [17]	2017	Netherlands	43	68 (52.6–81.1)	SBRT	[^18^F]Choline	PET/CT	baseline
Incerti et al. [15]	2017	Italy	20	67 (51–80)	sTomotherapy	[^11^C]Choline	PET/CT	baseline
Franzese et al. [18]	2017	Italy	26	71 (56–90)	SBRT	[^11^C]Choline	PET/CT	baseline and 6 months after therapy
Schlenter et al. [28]	2018	Germany	67	73 (51–83)	IMRT and SIB	[^18^F]Choline	PET/CT	baseline
Kalinauskaite et al. [19]	2020	Germany	50	62 (47–75)	SBRT	[^68^Ga]PSMA	PET/CT	baseline and restaging
Hurmuz et al. [20]	2020	Turkey	176	65 (42–84)	SBRT or IMRT	[^68^Ga]PSMA	PET/CT	baseline
Oehus et al. [14]	2020	Germany	78	64 (48–78)	SBRT or RT	[^68^Ga]PSMA	PET/CT	baseline and restaging
Onal et al. [21]	2021	Turkey	74	64 (49–87)	SBRT	[^68^Ga]PSMA	PET/CT	baseline
Bergamin et al. [31]	2020	Australia	25	72 (62–83)	SBRT	[^68^Ga]PSMA	PET/CT	baseline
Henkenberens et al. [22]	2020	Germany	32	63 (53–74)	RT	[^68^Ga]PSMA	PET/CT	baseline
Emmett el al., [10]	2020	Australia	129	68 (IQR 63–72)	IMRT	[^68^Ga]PSMA	PET/CT	baseline
Shakespeare et al. [29]	2019	Australia	46	70 (51–81)	SBRT	[^68^Ga]PSMA	PET/CT	baseline
Narang et al. [30]	2020	India	44	69 (52–84)	SBRT	[^68^Ga]PSMA	PET/CT	baseline
Baumann et al. [23]	2018	Germany	5	71(66–78)	IGRT	[^68^Ga]PSMA	PET/CT	baseline
Artigas et al. [24]	2019	Belgium	20	69 (56–82)	RT	[^68^Ga]PSMA	PET/CT	baseline
Guler et al. [25]	2018	Turkey-Belgium	23	68 (54–78)	IMRT-IGRT	[^68^Ga]PSMA	PET/CT	baseline
Emmett et al. [11]	2017	Australia	164	68 (IQR 62–71)	RT	[^68^Ga]PSMA	PET/CT	baseline
Schmidt-Hegemann et al. [12]	2018	Germany	129	72 (47–83)	RT	[^68^Ga]PSMA	PET/CT	baseline
Henkenberens et al. [26]	2017	Germany	42	66.5 (49–85)	RT	[^68^Ga]PSMA	PET/CT	baseline
Koerber et al. [27]	2021	Germany	86	65 (49–80)	RT	[^68^Ga]PSMA	PET/CT	baseline
Schwarzenböck et al. [33]	2016	Germany	32	70 (52–82)	docetaxel	[^11^C]Choline	PET/CT	baseline, after the 1st and 10th cyclee
Ceci et al. [35]	2016	Italy	61	67.5 (57–84)	docetaxel	[^11^C]Choline	PET/CT	Baseline, after 4 or 8 or 12 cycles
Schwarzenböck et al. [34]	2016	Germany	11	68.72 (5.83)	docetaxel + ADT	[^11^C]Choline	PET/CT	baseline and 7–21 days after therapy
Quaquarini et al. [36]	2019	Italy	29	71 (42–82)	docetaxel	[^18^F]Choline	PET/CT	baseline and 14–58 days after therapy
Seitz et al. [37]	2018	Germany	23	75 (54–82)	docetaxel	[^68^Ga]PSMA	PET/CT	Baseline and end of therapy (3 weeks after)
Anton et al. [38]	2020	Australia	25	63	docetaxel + ADT	[^68^Ga]PSMA	PET/CT	baseline and within 6 weeks following upfront docetaxel
Has Simsek et al. [39]	2021	Turkey	52	67 (53–82)	docetaxel	[^68^Ga]PSMA	PET/CT	at least after six cycles of therapy
Caffo et al. [40]	2014	Italy	12	77 (59–86)	enzalutamide	[^18^F]Choline	PET/CT	baseline, 3 and 6 months after therapy
Maines et al. [41]	2016	Italy	30	76.9 (59.2–86.7)	enzalutamide	[^18^F]Choline	PET/CT	baseline and during therapy
De Giorgi et al. [43]	2014	Italy	43	73 (57–87)	abiraterone	[^18^F]Choline	PET/CT	baseline and 3–6 weeks after the start of therapy
De Giorgi et al. [42]	2015	Italy	36	72 (48–90)	enzalutamide	[^18^F]Choline	PET/CT	baseline and 3–6 weeks after the start of therapy
Chen et al. [44]	2021	China	45	68 (62–72)	abiraterone	[^68^Ga]PSMA	PET/CT	before and after 6 months post-therapy
Zukotynski et al. [45]	2021	USA	16	71.5 (66.8–72.0)	abiraterone or enzalutamide	[^18^F]DCFPyL	PET/CT	at baseline and within 2–4 months after starting therapy.
Plouznikof et al. [46]	2019	Belgium	26	abiraterone 71.9 ± 7.6 enzalutamide 70.4 ± 9.6	abiraterone or enzalutamide	[^68^Ga]PSMA	PET/CT	Baseline and after 110 days (IQR 76–124) for enzalutamide and after 87 days (IQR 71–242) for abiraterone
Filippi et al. [47]	2020	Italy	20	75	[^223^Ra] dichloride	[^18^F]Choline	PET/CT	baseline, 1 month after 223Ra
García Vicente et al. [48]	2019	Spain	40	72.55 (±8.58)	[^223^Ra]dichloride	[^18^F]Choline	PET/CT	baseline
Ahmadzadehfar et al. [49]	2017	Germany	32	73 (62–84)	[^223^Ra]dichloride	[^68^Ga]PSMA	PET/CT	baseline and 4 weeks after full 223 Ra therapye

Legend: SBRT = stereotactic body radiation therapy; IMRT = intensity-modulated RT; SIB = simultaneous integrated boost; IGRT = image-guided RT; ADT = androgen-deprivation therapy; IQR = interquartile range; DCFPyL = PSMA agent.

## 4. Discussion

From the analysis of the recovered papers, some points of discussion emerged:

Firstly, a limited number of papers for radiolabeled choline are available against radiolabeled PSMA before RT, mainly for the higher detection rate of radiolabeled PSMA PET than radiolabeled choline in case of low PSA levels after RP. Similarly, also for the re-irradiation, radiolabeled PSMA seems to be preferred than radiolabeled choline PET. Conversely, for the evaluation of response to systemic therapies in mCRPCa, i.e., new hormonal therapies, docetaxel and [^223^Ra]dichloride, data about radiolabeled choline PET before and after therapy or only before were more often available than radiolabeled PSMA PET. Second, PSMA PET has an important impact in the prediction of survival outcomes, especially in patients with oligometastatic disease. Indeed, in patients with a limited extension of disease (a maximum of five lesions), PET with radiolabeled PSMA has an important impact on the biochemical free survival and ADT-free survival assessment. Therefore, patients with oligometastatic disease would have some survival beneficial from a baseline staging with radiolabeled PSMA PET. Secondly, we observed that only a few papers about the utility of [^68^Ga]PSMA-11 and [^18^F]Choline PET/CT for the assessment of response to RT are available. Data are scarce in this setting, and it would be interesting to assess whether the metabolic response assessed by molecular imaging has an impact on the local and distant control of the disease. Third, in most papers, a qualitative PET/CT assessment is the most common analysis, with limited use of semi-quantitative parameters for both PSMA and choline. Indeed, some clinical and methodological parameters can significantly affect the semiquantitative data from PET. Obviously, a visual or qualitative score would be more reproducible for the evaluation of response to systemic or local therapy in patients with recurrent PCa. Therefore, an important effort in this sense should be made. Lastly, the role of molecular imaging for the definition of response to chemotherapy or new hormonal agents, either as neoadjuvant or palliative–curative treatment, is still uncertain. Despite the paucity of data, it seems that the presence of a widespread disease (expressed as tumor burden or metabolic tumor activity) can be significantly linked to the prognosis. Indeed, some authors [45,46] observed that PSMA expression’s reduction on PET/CT were strong predictors of response to enzalutamide and abiraterone and it was associated with time to progression and OS. Therefore, the loss of expression of PSMA in patients with mCRPCa would be monitored if a therapy with 177-Lu PSMA is started.

To date, the number of therapeutic strategies is enough to delineate the correct indication for each of them. Therefore, well-designed studies with specific standardized protocols for the interpretation and the timing between therapies are mandatory. In our opinion, the extensive and wide introduction of 18F-labelled PSMA (i.e., [^18^F]DFCPyL and [^18^F]PSMA-1007) would have an important impact in this setting producing scientific evidences. Moreover, in the future, the introduction of PSMA-based radiopharmaceuticals labelled with 64Cu/67Cu in clinical practice would play a “true” theragnostic pairs.

As a final consideration, the present study has some limitations that emerged during the assessment of selected articles, such as: (1) different tomograph models (BGO vs.LSO/LYSO; analogic vs. digital PET) that can affect the lesion detection; (2) timing between PET and therapy; (3) clinical variability of patients with different initial stage and PSA values as well as the retrospective design of studies.

## 5. Conclusions

The advent of molecular imaging in the management of patients with PCa has a significant impact. Some indications have already been established in the main international guidelines, but some others are still uncertain. In patients undergoing RT, radiolabeled choline and [^68^Ga]PSMA-11 have an important prognostic role, also in postponing the ADT starting, thus reducing potential side effects. However, in case of systemic therapies, the role of new-generation imaging seems undervalued, thus requiring additional efforts in this way.

## Figures and Tables

**Figure 1 cancers-14-01770-f001:**
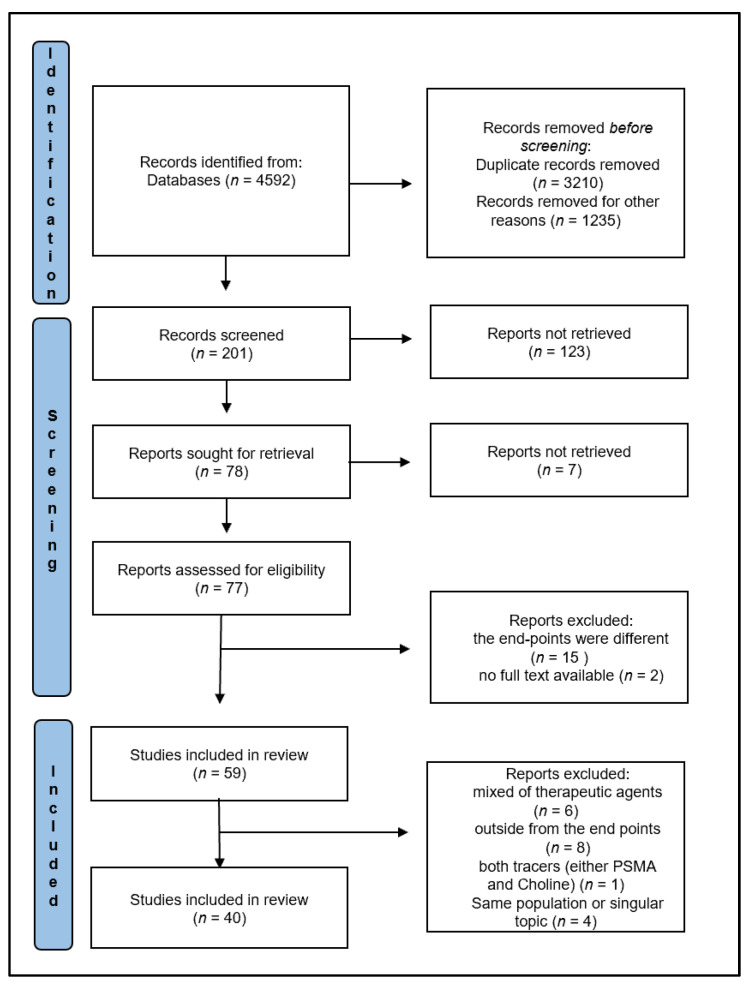
PRISMA flowchart.

**Figure 2 cancers-14-01770-f002:**
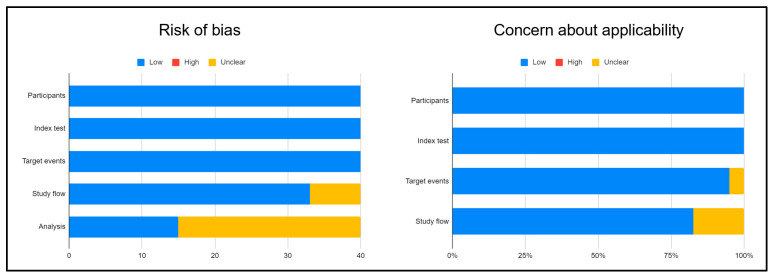
QUAPAS results for all selected studies.

**Figure 3 cancers-14-01770-f003:**
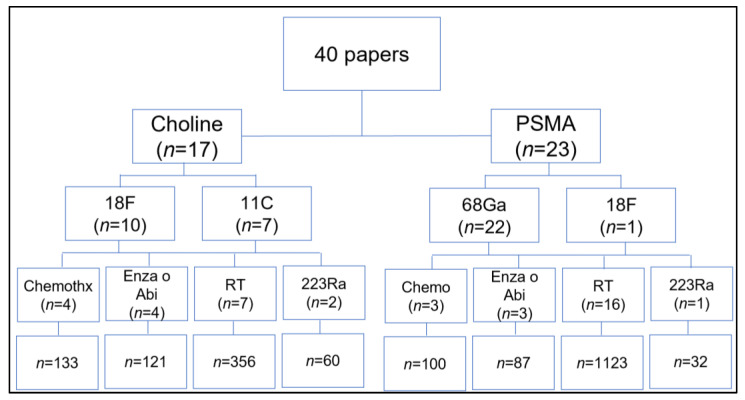
Distribution of the papers in accordance with their main endpoint. Chemo = chemotherapy, Enza = enzalutamide; Abi = abiraterone acetate; RT = radiotherapy; 223Ra = 223radium-dichloride.

**Table 2 cancers-14-01770-t002:** Summary of all 40 selected papers.

Authors, Ref	Type of PET Analysis	SoR	Agreement between PET and SoR	Cut-Offs	Median Follow-Up (Range)	Survival Analysis	Outcome
Incerti et al. [13]	qualitative and semiquantitaive (SUV_max_, MTV)	PSA	NA	MTV60 = 0.64	29 months (4–98)	MTV40, MTV50 and MTV60 correlated with OS; MTV60 correlated with recurrence	MTV60 and the presence of choline uptake outside the pelvic region were prognostic indicators of recurrence
Pasqualetti et al. [16]	qualitative	PSA	NA	NA	NA	systemic-therapy-free survival at 6, 12 and 24 months were 93.5%, 73.9%, 63.1%	[^18^F]Cho PET/CT can identify oligometastatic PCa suitable for SBRT, resulting in a systemic-therapy-FS of 39.1 months
Fodor et al. [9]	qualitative	PSA	NA	NA	36 months (9–116)	Extra- vs. intrapelvic LNs involvement and number of PET positive LNs predicts death	Extrapelvic positive LN location and number of positive LNs predicted the clinical relapse
Bouman-Wammes et al. [17]	qualitative	NA	NA	NA	2.6 years	Median ADT-free survival was 15.6 months for the whole group, and 25.7 months for patients with a PSA response	SBRT can safely and effectively be used to postpone ADT in patients with oligometastatic recurrence of PCa detected by PET/CT
Incerti et al. [15]	qualitative	PSA	NA	NA	2 years (1–7)	Complete or partial biochemical response occurred in 79% of patients, at 6 months in 82% and at 12 months in 63% of patients. bRFS at 2 years was 50%. OS at 2 years was 55%	CHO-PET/CT based HTT is a suitable therapeutic approach in patients with recurrent PCa presenting bone metastases
Franzese et al. [18]	qualitative (PERCIST)	PSA	NA	NA	29.4 months (2.9–79.5)	The 1- and 2-year PFS were 55.2% and 35.1%, respectively.The 1- and 2- year treated metastasis control was 79.6% and 74.9%, respectively.	PET is important in identification of gross tumor and evaluation of the response: median PSA-nadir after the end of the RT was 1.02 ng/mL
Schlenter et al. [28]	Semiquantitative (T/B choline uptake ratio > 2)	PSA	NA	NA	66 months	The 5-year overall survival was 100% in the SIB vs. 88% in the conventional group	The 5-year biochemical tumor control was 92% vs. 85% in the SIB vs. conventional groups
Kalinauskaite et al. [19]	Semiquantitative (SUVmedian 6)	NA	NA	NA	34 months (5–70)	Median PFS was 12 months (range: 2–63) and TFFS was 14 months (range: 2–70). 64% patients had repeat oligometastatic disease. 48% patients with progression underwent a second SBRT course. Two-year LC after SFRS was 96%.	For patients with OMPCa, PSMA-guided SBRT might be used as an alternative to ADT, deferring the start/ escalation of palliative ADT and its side effects. Metastases treated with PSMA-PET/CT-based SFRS reached excellent LC with minimal toxicity
Hurmuz et al. [20]	qualitative	NA	NA	NA	22.9 months	The 2-year OS rate was 87.6% and the median OS was not reached. The 2-year PFS rate was 63.1% and median PFS was 39.3 months. At a median of 17.9 months (2.1–24.3 months) after completion of MDT, 9 patients (0.5%) had local recurrence at the oligometastatic site. The 1- and 2-year LC rates at the treated oligometastatic site per patient were 98.1% and 93.2%, respectively.	[^68^Ga]PSMA-11 PET/CT-based MDT leads to excellent local control rates. Together with effective systemic management, it has the potential to contribute to achieving long-term survival in selected patients
Oehus et al. [14]	qualitative	PSA	NA	NA	16 months (3–54)	OS was 97.4% after 2 years. Median bRFS was 17.0 months (14.2–19.8). After 12 months, 55.3% were free of biochemical progression. Concurrent ADT was the most important independent factor for bRFS (*p* = 0.01). Exploratory statistical analyses estimated a median ADT- FS of 34.0 months.	PSMA - PET-based RT for recurrent PCa with limited tumor burden was effective and safe
Onal et al. [21]	qualitative	PSA	NA	NA	27.3 months	The 2-year PCSS and PFS rates were 92.0% and 72.0%, respectively.	SBRT is an efficient and well-toleratedtreatment option for PCa patients with 5 bone-only oligometastases or fewer detected with [^68^Ga]PSMA-11 PET/CT
Bergamin et al. [31]	Semiquantitavie (MTV with MRI delineation of GTV)	Histopathology	100%	NA	25 months	Twenty-four patients had a 12-months [^68^Ga]PSMA-11 PET/CT, 23 (92%)with no evidence of disease. Ten of the 12 patients with routine 24-month [^68^Ga]PSMA-11 PET/CTs were disease free. The 2-year bFFF estimated was 79.6%.	PSMA-directed salvage focal reirradiation to the prostate using linear accelerator based SBRT is feasible and safe
Henkenberens et al. [22]	qualitative	PSA	NA	not for PET	39.5 months (18–60)	Median bPFS_1 was 16.0 months after the first PSMA PET-directed RT and the median bPFS_2 was 8.0 months after the second PSMA PET-directed RT. Median ADT-FS was 31.0 months	Repeated PSMA PET-directed RT for oligorecurrent PCa postponed ADT without significant toxicities
Emmet et al. [10]	qualitative and semiquantitative(SUV_max_)	PSA	NA	NA	38 months (IQR 31–43)	FFP at 3 years in 64.5% of men who underwent sRT. 3 years FFP was 81% in those with negative or fossa-confined findings; 45% in PSMA PET–positive disease outside the prostatic fossa.	PSMA PET results are highly predictive of FFP at 3 years in men undergoing sRT for BCR after RP Negative PSMA PET results or disease identified as still confined to the prostatic fossa demonstrated high FFP
Shakespeare et al. [29]	NA	NA	NA	NA	24 months (10–50)	Two-year failure-free survival was 100%, and two-year overall survival 95.7%	PSMA PET-guided IMRT up to 81 Gy to the prostate and involved LNs, and long term ADT, is a promising approach for newly diagnosed LN positive PCa
Narang et al. [30]	GTV	NA	NA		63.5 months (17–92)	5 years PFS was 88.2%, biochemical PFS was 91.4% and OS was 96.9%	Excellent five-year outcomes can be obtained even for locally advanced, node-positive and bone oligometastatic PCa, by means of dose-escalation using EH-SBRT boost to the prostate
Baumann et al. [23]	qualitative PERCIST	PSA	NA	reduction in SUV_max_ of at least 30% with no increase in lesion size.	11 months (2–15)	PFSl of 88% after 1 year	PSMA PET-detected metastatic lesions can be effectively treated with high-precision radiotherapy to the PSMA PET-positive tumor volume
Artigas et al. [24]	qualitative	PSA	NA	NA	15 months (4–33)	BCR-free survival rate at 1 year was 79% and 53% at 2 years. ADT-free survival at 2 years was 74%.	Metastasis-directed RT based on [^68^Ga]PSMA-11 PET/CT may be a valuable treatment in patients with PCa oligometastatic disease, providing promising BCR-free survival rates and potentially postponing ADT for at least 2 years in 74% of the patients
Guler et al. [25]	Semiquantitative (SUV_max_)	PSA	NA.	NA	7 months (2–17)	The actuarial 1-year LC, PFS and OS rates were 100, 51 (95% CI 8–83%) and 100%.	PSMA PET-CT-guided RT may be an attractive treatment strategy in patients with oligo-metastatic PCa providing optimal LC, low toxicity and a promising PFS
Emmet et al. [11]	qualitative and semiquantitative(SUV_max_)	PSA	50%	NA	10.5 months (IQR 6–14)	Overall treatment response after SRT was 72%. 45% of patients with a negative PSMA underwent SRT whereas 55% did not. In men with a negative PSMA who received SRT, 85% demonstrated a treatment response, compared with a further PSA increase in 65% in those not treated.	PSMA PET is independently predictive of treatment response to SRT and stratifies men into a high treatment response to SRT (negative or fossa-confined PSMA) versus men with poor response to SRT (nodes or distant-disease PSMA)
Schmidt-Hegemann et al. [12]	qualitative	PSA	74%	NA	20 months (3–42)	94% of patients with biochemical recurrence and 82% of patients with biochemical persistence having a PSA ≤ 0.2 ng/ml.	PSMA PET/CT-based radiotherapy is an effective local salvage treatment option with significant PSA response in patients with biochemical recurrence or persistence after radical prostatectomy leading to deferral of long-term ADT or systemic therapy
Henkenberens et al. [26]	qualitative - CTV PET based	PSA	NA	NA	39 months	The median bPFS was 12.0 months after PSMA ligand PET-based RT and the median SST-FS was 15.0 months.	PSMA PET-guided RT represents a viable treatment option for patients with oligometastatic mCRPCa to delay further systemic therapies
Koerber et al. [27]	qualitative and semiquantitative(SUV_max_)	PSA	83.1%	NA	26 months	3-years OS and bPFS were 84% and 55%, respectively. The median time of ADT-free survival was 13.5 months	PSMA-guided RT is a promising therapeutic approach in guiding to RT in OMD
Schwarzenbock et al. [33]	Semiquantitative (SUV_max_ and SUV_mean_)	RECIST	30%	NA	NA	NA	No correlation among PSA, RECIST and SUVs by choline PET
Ceci et al. [35]	EORTC criteria	PSA	47.5% disagreement	none	13.5 months (6–53)	increasing PSA trend after docetaxel predicts PD	PET/CT identified PD despite PSA response. Tumor burden at baseline predicted PD
Schwarzenböck et al. [34]	semiquantitative	Histopathology	91%	none	NA	NA	Decrease in choline uptake after combined neoadjuvant therapy are paralleled by regressive and apoptotic changes in histopathology
Quaquarini et al. [36]	Semiquantitative (SUV_max_, SUV_mean_, PVC-SUV, MATV, TLA, PVC-TLA). Whole-body indices	NA	NA	SMATV value > 27 cc	6 years	SMATV, and STLA had a statistically significant correlation with PFS	Semi-quantitative indexes such as SMATV and STLA at baseline have a prognostic role in patients treated with docetaxel
Seitz et al. [37]	qualitative (PERCIST)	CT	50–86%	NA	NA	NA	[^68^Ga]PSMA-11 PET is a promising tool for the assessment of response to therapy
Anton et al. [38]	qualitative and semiquantitative (SUV_max_, MTV, total lesional PSMA-expression)	PSA, CT	100%	NA	37.7 months	NA	PSMA PET/CT response by visual analysis is concordant with the established good prognostic marker of PSA ≤ 0.2 ng/mL following chemo-hormonal therapy, and may be a better predictor of CRPCa
Has Simsek et al. [39]	Semiquantitative (SUV_max_, SUV_mean_, TV-PSMA, TL-PSMA)	PSA	sensitivity: 75% and specificity 80%	TV-PSMA 107 cm^3^ and TL-PSMA 1013 cm^3^	60 months	The median TTP was 16 months, and the median OS was not reached. High TV-PSMA, highTL-PSMA, high age, and high LDH were associated with shorter OS, while high TV-PSMA and high age were significantly related with shorter TTP.	Patients with high TV-PSMA had a significantly higher risk for chemotherapeutic failure. PSMA-based tumor burden prior chemotherapy seems to be a reliable predictive tool for survival in mCRPCa patients.
Caffo et al. [40]	EORTC criteria	PSA	50%	NA	NA	no	FCH PET/CT could assess the effect of enzalutamide in extensive PCa disease
Maines et al. [41]	EORTC criteria	PSA	20/30	NA	16.5 months (2.9–18.9)	baseline SUV_max_ was correlated with BCR	baseline PET/CT can be useful to define patients who will benefit from enzalutamide after docetaxel
De Giorgi et al. [43]	qualitative	PSA	71%	NA	22 months	progression PET, PFS: 3.4 months non-progression PET, PFS: 12.8 months	Association of FCH PET and PSA can improve the prediction of outcome and to monitor the response to therapy in mCRPCa
De Giorgi et al. [42]	qualitative	PSA	71%	NA	24.2 months (1.8–27.3)	PET/CT (DC or PD) predicted only PFS	Association of FCH PET and PSA can improve the prediction of outcome and to monitor response to therapy in mCRPCa
Chen et al. [44]	semiquantitative (SUV_max_ and MTV)	Histopathology	Specificity: 89%	NA	NA	NA	[68Ga]PSMA-11 PET/CT has a better diagnostic performance of pathological response to neoadjuvant treatment compared with PSA
Zukotynski et al. [45]	semiquantitative (SUV_max_, DPSM, DASM)	Conventional imaging and RECIST 1.1	NA	NA	28.2 months	low DPSM had median TTTC 12.2 months and median OS 37.2 months; high DPSM had median TTTC 6.5 months and median OS 17.8 months. Low DASM had median TTTC 12.2 months and median OS. High DASM had median TTTC 6.9 months and median OS 17.8 months.	Findings on PSMA-targeted PET 2–4 months after initiation of abiraterone or enzalutamide is associated with TTTC and OS
Plouznikof et al. [46]	Dominant response criteria, SUVmax	PSA	NA.	NA	3 months	NA	PSMA PET/CT response evaluation is strongly associated with response to treatment in mCRPCa patients under enzalutamide or abiraterone
Filippi et al. [47]	qualitative and TLA variations analysis	clinical, laboratory, imaging	100%	50% TLA (TLG) reduction	NA	median OS TLA-responder 19 vs. 8 months TLA-non-responders	Reduction in [^18^F]Cho PET/CT TLA after 223Ra seems to be related to longer survival
García Vicente et al. [48]	qualitative and semiquantitative (SUV _max_, average SUVmax for the five referred lesions)	PSA	NA	NA	NA	The extension of the bone disease by FCH PET/CT, SUV_max_ and average SUV_max_ were related to OS. No significant association was found for the PFS.	FCH PET/CT had a prognostic role in the prediction of OS. None clinical or imaging variable were able to predict the PFS
Ahmadzadehfar et al. [49]	qualitative	Bone scan, PSA, ALP	Significant correlation	NA	NA	NA	Improved therapeutic response in patients who underwent PSMA PET/CT as a gatekeeper for 223Ra therapy, because of better patient selection, mainly with the exclusion of patients with bone marrow involvement

NA = not available; MTV = metabolic tumor volume; OS = overall survival; FS = free survival; SBRT: stereotactic body radiation therapy; LN = lymph node; ADT = androgen deprivation therapy; PCa = prostate cancer; bRFS = biochemical recurrence free survival; HTT = hormonal therapy time; RT = radiation therapy; SIB = simultaneous integrated boost;T/B = tumor to background, PFS = progression free survival; TFFS = time to free survival; LC = local control; SFRS = single fraction radio-surgery; OMPC = oligometastatic prostate cancer; PD = progressive disease; PCSS = prostate cancer specific survival; MRI = magnetic resonance imaging; GTV = gross tumor volume; mCRPC = metastatic castrate resistant prostate cancer; FFP = free from progression; PVC = partial volume effect; MATV = metabolic activity tumor volume; TLA = total lesion activity; TL = total lesion; TV = total volume; DPSM = delta percentage of SUVmax; DASM = delta absolute of SUVmax.

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
