# Peer review of "PSMA and Choline PET for the Assessment of Response to Therapy and Survival Outcomes in Prostate Cancer Patients: A Systematic Review from the Literature"

_cancers, 2022, doi:10.3390/cancers14071770_

Round 1
Reviewer 1 Report
The review compares PSMA and Choline/PET as prognostic biomarkers in PCa. The review is well organized and can be useful to clinical colleagues in the field.
It's a nice review on how PSMA and CH PET can assess response to therapy. PSMA PET is a very hot topic these days, and review material is useful for people in the field.
Only the English isn’t excellent in the second half of the review and the author should follow the right nomenclature for radiopharms and make it homogeneous throughout the manuscript.
Please obey the consensus nomenclature rules for radiopharmaceuticals (https://www.sciencedirect.com/science/article/pii/S0969805117303189 and follow-up papers). The discussion and conclusion will benefit from editing done by a native English speaker.
Author Response
Thank you for these suggestions. We now extensively improved the manuscript’s English and radiopharmaceuticals’ nomenclature accordingly.
Reviewer 2 Report
The authors shoud be congratulated about the extensvise search and description of the different PET studies. They describe 40 studies that were published in the last decade conceirning prostate cancer and PET imaging in relation to therapeutic outcome.
However, I do have some conceirns about the heterogeinity of the included papers. Both the type of PET scans vary as are the type of prostate cancer patients ans their subsequent treatments. 17 out of 40 studies report about choline. Choline PET scanning is inferior to PSMA PET and this makes results of these studies less relevant. I would recommend to discuss this topic more profoundly.
The paper (including the title) focusses on response. But in most cases you mean prediction and/or prognosis. The term response makes it sometimes unclear. I think the term response should be reserved for those studies that perform imaging pre and post-treatment. Than you truly assess PET response (changes) and relate this to outcome parameters.
Author Response
Response 1. Dear Reviewer, all the articles evaluated in this Review regards PCa (prostate neuroendocrine tumors were not considered). The aim of the study was stated to assess the utility of PSMA-PET and Choline-PET in the assessment of response to systemic and local therapy. We think the current management of PCa still regards both radiotracers despite the developing role of PSMA-PET that could replace Choline-PET in some clinical conditions. Despite the slight minor number of papers on Choline-PET, we preferred to keep all contributions for the interest for the worldwide readers. Indeed, and unfortunately, PSMA-PET is still experimental or difficult to produce in some realities. Moreover, the objective of the study did not regard the comparison of PSMA vs Choline, but to assess the utility of both tracers in some specific clinical needs, such as the survival outcomes and the prediction of response to therapy. Nevertheless, we have included more sentences in the discussion paragraph, mainly comparing the available data with Choline and those that have been collected with PSMA.
However, in order to complete the discussion paragraph, we added the limitations of the study considering some of your comments as follow: “As final consideration, the present study has some limitations that emerged during the assessment of selected articles, such as: 1) different tomograph models (BGO vs.LSO/LYSO; Analogic vs Digital-PET) that can affect the lesion detection; 2) timing between PET and therapy; 3) clinical variability of patients with different initial stage and PSA values as well as the retrospective design of studies. Response 2. We agree with the reviewer about the limited significance of term “response”. However, we preferred to extend our study to paper evaluating the predictive and prognostic role of PET imaging in this field as directly correlated to therapy responses. To meet your suggestion, we changed the title as follow: PSMA and Choline PET for the Assessment of Response to Therapy and Survival Outcomes in Prostate Cancer Patients: A Systematic Review from the Literature.
